# A Review of Recent Advances in Nanoengineered Polymer Composites

**DOI:** 10.3390/polym11040644

**Published:** 2019-04-09

**Authors:** Vishnu Vijay Kumar, G. Balaganesan, Jeremy Kong Yoong Lee, Rasoul Esmaeely Neisiany, S. Surendran, Seeram Ramakrishna

**Affiliations:** 1Centre for Nanofibers and Nanotechnology, Department of Mechanical Engineering, National University of Singapore, Singapore 117575, Singapore; vishnuv36@gmail.com (V.V.K.); jeremy.lee@lr.org (J.K.Y.L.); 2Department of Ocean Engineering, Indian Institute of Technology, Madras 600036, India; sur@iitm.ac.in; 3Department of Mechanical Engineering, Indian Institute of Technology, Jammu 181221, India; balaganesan.gurusamy@iitjammu.ac.in; 4Division of Materials Science, Luleå University of Technology, SE-97187 Luleå, Sweden; rasoul.esmaeely@ltu.se

**Keywords:** polymer composites, impact, nanofiber interleave, interfacial toughening, self-healing composites

## Abstract

This review paper initially summarizes the latest developments in impact testing on polymer matrix composites collating the various analytical, numerical, and experimental studies performed since the year 2000. Subsequently, the scientific literature investigating nanofiller reinforced polymer composite matrices as well as self-healing polymer matrix composites by incorporating core-shell nanofibers is reviewed in-depth to provide a perspective on some novel advances in nanotechnology that have led to composite developments. Through this review, researchers can gain a representative idea of the state of the art in nanotechnology for polymer matrix composite engineering, providing a platform for further study of this increasingly industrially significant material, and to address the challenges in developing the next generation of advanced, high-performance materials.

## 1. Introduction

Laminated polymer matrix composites are well-known materials for diverse applications. They have fibre reinforcements ingrained in a matrix of either thermoplastic or thermosetting polymers. The fibre diameters may range from a few millimeters to the sub-nanometer range [1]. The advantages of fibre reinforced composites include high specific strength and modulus, facile fabrication, high design flexibility, good resistance to fatigue and corrosion, desirable thermal expansion characteristics, and economic efficiency [2]. These properties make them suitable for a wide and diverse array of applications in all major fields, such as aerospace [3], automobile [4], construction [5], communication [6], orthopedics [7], dentistry [8], energy [9], military [10], and several other exotic, high-performance engineering applications [2]. There has been considerable research aimed at further enhancing the properties of these polymer matrix composites by the addition of materials, like nanofillers, nanofibers, etc. [11]. The dynamic response of these materials under various loading conditions was studied to check their suitability for certain applications [12,13]. One such dynamic response study is the impact analysis of composite materials, which provides useful information on the behavior of composites under sudden load applications. This paper discusses the addition of nanofibers and a nanofiber based self-healing approach in this class of nanofiber interleaved composites, briefly describing the improvement of properties occurring in polymer composites by the addition of nanoscale particles/fibers, i.e., nanoengineering polymer composites.

The paper is basically divided into four major sections pertaining to the response of composite materials to impact loading. The first section deals with the basic study of impact on composite materials, including a brief review of the various historical events associated with high-velocity impact studies. The second section discusses the use of electrospun nanofiber interleaved polymer matrix composites and various property enhancements achieved through this technique. The third section deals with the addition of nanofillers into composites and their response to impact loading. Finally, the fourth section details the self-healing of polymer matrix composites using core-shell nanofibers.

### 1.1. Impact on Composites

Impact studies on composite laminates have always been an area of concern for various researchers globally [14,15,16,17,18,19,20,21,22]. Impact is defined as the collision between two or more bodies, where the interaction between the bodies can be elastic, plastic, fluid, or any combination of these [23]. In general, there are five types of velocity considered for the study of impact responses: Low velocity [24,25], sub-ordnance [26], ordnance, ultra-ordnance [27], and hypervelocity [28]. The characteristic testing methods and typical applications of each impact velocity regime are shown in Table 1 [29]. Among these, high-velocity impact studies happening in the subordinate range are an area of particular concern.

The ballistic limit velocity is a benchmark for measuring the material’s nature in withstanding a projectile impact. It is calculated as an average of the maximum velocity that does not penetrate and the minimum velocity leading to perforation.

During a ballistic impact, the energy of the projectile is absorbed by the target through [23]:Moving cone formed on the back face of the target.Shear plugging of the projectile into the target.Tensile failure of the primary fibers.Elastic deformation of the secondary fibers.Matrix cracking and delamination and frictional energy absorbed during penetration.

Under high-velocity impact conditions, most of the available energy is dissipated over a small zone near the point of contact [30]. The best way to analyze the impact events are by studying the two characteristic parameters, namely the impact damage resistance, and the impact damage tolerance. The former is the response and damage of the structures caused by an impact; whereas the latter deals with the effect of existing impact damage on the strength and stability of the structures [31]. The parameters that govern the nature of high-velocity impact events on polymer matrix composites include the material, laminate thickness, architecture and volume fraction of the reinforcement, projectile geometry, matrix system, and mass [32]. Impact on composite structures sometimes results in delamination cones through the thickness, which irreversibly damages the structure [33,34]. Enhancing the ballistic resistance of armors by using composite materials is a generally accepted practice in armor design. In the past few decades, laminate armors have been investigated and found to achieve good durability and lightweight properties [35].

### 1.2. State-Of-The-Art in Composite Impact Research

An extensive literature review has been carried out in the field of high-velocity impact events using the Scopus database, from the year, 2000, through to 2018 [15,36,37,38,39]. The sheer number of studies performed clearly evidences how the area has grown over the years, with the level of sophistication evolving from using simple fiber sandwich panels reinforced with natural fibers, to the present use of hybrid nanoengineered interleavings over a short period of time.

## 2. Electrospun Nanofiber Interleaved Polymer Matrix Composites

There is no doubt that composites are a class of materials receiving much attention world-wide as an alternative for various existing structural and lightweight, high strength materials. The synergistic properties of their components make them superior in numerous applications due to advantages, such as high specific strength and modulus, facile fabrication, high design flexibility, good resistance to fatigue and corrosion, desirable thermal expansion characteristics, and economic efficiency [40,41,42]. These properties can further be tailored to suit certain intended functions via control over the composition and fabrication of the composite. One such method is the interleaving of nanofibrous mats, which are prepared via various techniques, like drawing, electrospinning, self-assembly, template synthesis, and thermal-induced phase separation [43]. Figure 1 schematically shows a laminated composite with and without interleaved nanofibers [42], while the Field Emission Scanning Electron Microscope (FESEM) images present the effect of an incorporation of the nanofibers on the failure surface roughness of the composites [41]. It can be seen that after incorporation of the nanofibers, the brittle failure of the composite’s interlayer has transitioned to tough failure.

### 2.1. Electrospun Nanofiber Mats

Electrospinning is one of the most commonly used methods to produce nanofibers ranging from several micrometers to the sub nanometric ranges [1]. Nanofibers are a suitable candidate for interleaving owing to their low thickness, porosity, high volume efficiency, less density, superior mechanical properties, etc. [44]. Electrospinning is a highly accepted method as it generates nanofibers from a straightforward, scalable setup for efficient mass production. Furthermore, electrospun nanofibers are ultrathin fibers with controllable diameters, compositions, and orientations [45]. Electrospun nanofibers are thus inexpensive, continuous, and relatively easy to align, assemble, and process in applications [46]. Nanofibrous mats are an ideal reinforcement to be interleaved between two plies of resin matrix composites [47,48] and adhesives [49,50,51] because of the following aspects [34]:Thinness and lightness: They can range from few microns to even subnanometric ranges and are as light as a few grams per square meter, making the impact to weight and thickness of the final manufactured negligible.Porosity: They are highly porous, allowing the matrix to impregnate through them while still maintaining a solid bonding between the two layers of fibers encapsulating them.Tiny volume: The actual volume of these nanomats is negligible owing to the fact that they accumulate with the resin.Mechanical properties: They show highly improved properties compared to their bulk counterparts.

There have been property improvements for interleaved nanofibers of various materials, with Table 2 showing the improvements derived from only nylon nanofiber interleaving for illustrative purposes.

### 2.2. Impact on Nano-Interleaved Composites

Impact on nanofiber interleaved polymer matrix composites has also been studied. Due to high-velocity impacts, a cone shape forms on the composites. Depending on the nature of the impact, such a cone can form from the top to the bottom of the laminate (⋁ shaped), or from the bottom to the top (⋀ shaped). The main aim of interleaving is to prevent the formation of such cones and to strengthen the structure. More research in the field is required to study these mechanisms and develop analytical models on the impact events.

## 3. Nanofiller Reinforced Polymer Matrix Composites

The dynamic response and energy absorption characteristics of composite laminates fabricated from glass fiber/epoxy composites with and without nanofillers in quasi-static and impact loading have been studied by many researchers [53,54,55,56,57,58]. The matrix is prepared with epoxy and nanoclay of 1–5 wt.% to understand the effect of the filler on the basic mechanical properties. In their work, the nanocomposite laminates were prepared by hand lay-up and compression molding processes. An impact testing set-up was also developed to conduct projectile impact testing for velocities up to 225 m/s for a 7.6 g mass projectile, which is used in this study. During this testing, he laminate edges are clamped as boundary conditions to fix the target. The un-impacted and post-impacted laminates are tested to predict vibration parameters, such as natural frequency (NF) and damping factors (DF). The effect of filler dispersion is verified by comparing NF and DF values of specimens with and without fillers of post impacted and un impacted specimens. These researchers also presented the delamination area of the specimens subjected to below and above ballistic velocities. A quasi-static penetration test was also conducted to identify the perforation resistance of the specimens with the same boundary condition.

Balaganesan et al. conducted experiments to identify the quasi-static properties of nanocomposite specimens. The tensile and flexural properties were predicted for the specimens with and without filler dispersion. They observed that filler dispersion up to 4% in the matrix had a uniform, exfoliated structure. It was also observed that the specimens of epoxy with 3% clay showed enhancements in tensile strength, tensile modulus, flexural strength, flexural modulus, and impact strength when compared to neat epoxy specimens of 13%, 17% and 15%, 11%, and 22%, respectively. They also prepared specimens with glass fiber as the primary reinforcement and nanofillers as the secondary reinforcement. The specimens of glass/epoxy with 3% clay showed an enhancement in tensile strength, which was 32% higher when compared to neat epoxy specimens. The authors conducted a test to find the strain energy release rate, G_IIC_, on glass/epoxy specimens using the end notched flexural test and they observed that laminates with clay up to 5% showed a higher energy release rate than neat epoxy specimens. The enhancement in the strain energy release rate for the glass/epoxy with 5% clay was about 25% when compared to the neat epoxy specimen.

The authors prepared glass/epoxy laminates of three, five, and eight layers with and without nano fillers as a secondary reinforcement to predict the perforation resistance under a quasi static punch test. It was observed that the specimens with clay up to 5% offered higher resistance when compared to neat epoxy specimens. The enhancement in energy absorption was 60% for three layer laminate with 5% clay when compared to neat epoxy specimens.

Velmurugan and Balaganesan used modal analysis on pre-impacted and post impacted nanocomposite specimens. The NF and DF were predicted using the impulse hammer technique for un-impacted and post-impacted laminates, with all the edges under clamped boundary conditions.

The NF values and DF values were obtained for the first five modes. The change in NF for three, five, and eight-layer nanocomposite laminates was observed in all the modes due to the presence of nano clay up to 5% by weight in all the modes. The frequency response function (FRF) plot for the nanocomposite specimen with 3% clay is shown in Figure 2. The change in NF for all the modes was more than 10% higher when compared to the neat epoxy specimen. The change in mode 1 NF was 11.4% higher when compared to the neat epoxy specimen. The change in NF of the three-layer laminate with 5% clay was 17.7% higher than the neat epoxy specimen in modes 1 to 5. The DF value of the specimen with 3% clay was higher than the other specimens. The authors observed the same trend in all the modes of three, five, and eight-layer laminates. The enhancement of the DF for the three-layer laminate with 3% clay was 73% higher than the neat epoxy specimen in mode 1. The NF values in mode 1 for the impacted laminates were about 10% less than that in un-impacted laminates, which was due to the damage caused by the impact loading.

The damage of the laminate due to fiber failure and delamination changes the vibration parameters, NF and DF, of the specimens. The authors observed that the DF values were high in post-impacted specimens, which was due to the interaction between layers separated in delamination. For post-impacted specimens, the NF decreased, which was due to a decrease in the stiffness, and DF increased with an increase in the impact velocity.

The authors also conducted an elaborate study on damage scenarios between specimens with and without filler. They observed a decrease in the delamination area for specimens with filler when compared to specimens without filler during impact loading below ballistic velocities. This was due to the nano filler, which acts as a secondary reinforcement and supports better bonding between layers. It clearly indicates that as the impact velocity increases, the delamination area increases, due to which the NF values of post-impacted laminates decrease. However, in nano-filled composites, the decrease in NF is less as the filler reduces the area of delamination.

Balaganesan et al. worked on the dynamic response of nanocomposite laminates during impact loading. They used a shock accelerometer with a 100 kg value. They obtained an acceleration-time response and frequency response of the specimens with and without nano fillers. The time response curve was plotted between the acceleration and the period of vibration for different impact velocities. The frequency response was plotted for NF values in various vibration modes of the laminates. The acceleration-time response for the nanocomposite specimen with 3% clay based on the authors’ research work is shown Figure 3, where the specimens were subjected to velocities below and above the ballistic limit.

They observed that the maximum amplitude of acceleration for impact velocities that are below the ballistic velocity was 10 times higher than that for velocities that are above the ballistic limit. In this particular case, for specimens with 3% filler, the maximum amplitude of acceleration when subjected to an impact velocity of 35 m/s was 48,522 m/s^2^. The maximum amplitude of acceleration of the specimen with the same configuration when subjected to an impact velocity of 135 m/s was 4892 m/s^2^.

It was also observed in another work [54] that the amplitude of acceleration for nanocomposite specimens was less than specimens without clay when subjected to similar impact velocities. In the same research work, it was observed that the maximum amplitude of acceleration for the three-layer specimen with 3% clay was 36% less than the neat epoxy specimen. Another interesting observation made in their study was on the total period of oscillations. They considered the accelerometer response as specimens’ oscillation cycles. It was observed that the period of oscillations for below ballistic impact velocities was higher than above ballistic velocities. Also, the specimen thickness changed the number of oscillations. Specimens with higher thicknesses and oscillations had higher numbers of cycles.

The authors also observed that specimens of fewer thickness values dissipated energy at higher modes of vibration. This was observed by the authors from the signal of the accelerometer. In their study, they conducted impact tests on three, five, and eight layer glass/epoxy laminates with and without nano fillers. Three and five layer specimens showed higher amplitude values at higher NF values and the eight layer specimen showed the same amplitude value in all the modes of vibration. Balaganesan and Velmurugan [54] studied the effect of clay dispersion on glass/epoxy specimens when subjected to impact velocities at below the ballistic limit. At these velocities, there was no damage to the fibers and projectile rebounds. The strain energy absorbed by the specimen was dissipated in the form of vibration. The failure modes of the specimens changed when the impact velocities were above the ballistic limit. The additional failure mode was a failure of the fibers due to perforation of the projectile. The fibers that are directly in contact with a projectile during perforation are called primary fibers and fibers that are not in contact with the projectile are called secondary fibers. The failure modes are the elastic deformation in the direction of the projectile movement, damage due to delamination, epoxy matrix cracks, and fracture of fibers.

The conclusions from the study [58] of energy dissipation by nanocomposite specimens subjected to impact velocities below the ballistic limit were as follows.

The authors presented that the kinetic energy of the projectile is absorbed by the specimens in the form of deformation under strain and then vibration, failure due to delamination, and matrix cracking. It was also observed that the energy absorbed by the laminate in vibration is the energy absorption in vibration for impact velocities that are below the ballistic limit, but higher than the energy absorbed due to delamination and matrix cracking. The presence of clay in the matrix enhanced the interaction between the clay and polymer within the elastic limit, which reflects an increase in energy absorption due to vibration. It was observed that the specimen with 3% clay showed a higher increase in energy absorption during vibration when compared to the neat epoxy specimen, which is due to the exfoliated structure formation within the matrix.

The ballistic limit of a three-layer glass/epoxy laminate is about 100 m/s. The authors conducted an impact test at the velocity of 82 m/s in a three-layer glass/epoxy specimen and the energy absorption in various failure modes is shown in Figure 4. The projectile energy for an 82 m/s impact velocity is 25.55 J. The energy absorption by the glass/neat epoxy specimen due to vibration is 6.46 J. The specimen with 3% clay absorbs 9.61 J of energy in vibration, which is 50% higher than the specimen without clay. It was observed that energy absorption due to delamination and epoxy matrix cracking is less than the energy absorbed during vibration for the neat epoxy specimens and specimens with clay. The delamination was suppressed in nanocomposite specimens. It was observed that the delamination energy of specimens with 3% clay is less than 50% of the specimen without clay. The matrix crack energy for the specimen with 3% is about 50% of the specimen without clay. The three-layer specimen with 3% clay when subjected to 82 m/s absorbs 50% more energy during vibration than that in the specimen without clay.

It was observed by the authors that clay dispersion in the matrix decreases the energy absorbing capability of specimens with clay up to 5% in the failure modes of delamination and matrix cracking for below ballistic impact velocities [59]. A three-layer specimen with 5% clay absorbed 50% less energy due to damage from delamination and matrix cracking than the neat epoxy specimen for an impact velocity of 82 m/s, which was also less than the perforation velocity.

Velmurugan and Balaganesan studied the ballistic limit and energy absorbed by nanocomposite specimens subjected to the impact of a target at above ballistic velocities. Fibers that are in direct contact with the projectile are termed primary fibers and those that are not in contact are termed secondary fibers. Secondary fibers were subjected to elastic strain energy in the deformation mode in the direction of the projectile movement. In this case, the kinetic energy of the projectile before striking the target specimen was dissipated by the failure of the specimens in elastic deformation of the secondary fibers, delamination, matrix cracking, and tensile failure of the primary fibers.

When the laminates were subjected to an impact velocity above the ballistic limit, the presence of clay in the epoxy matrix enhanced the impact resistance of the nanocomposites when compared to neat epoxy specimens [60]. The enhancement in the ballistic velocity was observed for the specimen with clay up to 5%. The ballistic velocity of the three-layer specimen with 5% clay was 27% higher than the neat epoxy specimen. The energy absorbing capacity of the nanocomposites was higher than the specimen without clay. The enhancement in the total energy absorption in various failure modes was observed for a specimen with a clay dispersion of up to 5%. For a three-layer specimen with 5% clay, when subjected to an impact velocity of 135 m/s, the enhancement of the energy absorption was 50% higher than that of the target specimen without clay. The damage due to delamination was high for nanocomposites when specimens were subjected to the above ballistic impact velocities. The area of delamination for the three-layer specimen with 5% clay was 87% higher than the specimen without clay during impact at 135 m/s.

The authors conducted experiments for various initial velocities of the projectile and obtained residual velocities for each case of the initial velocity. The authors also considered various other impact studies [61,62,63] and the results of the initial projectile velocity and projectile velocity after penetration of the three-layer specimens with clay and without clay are shown in Figure 5. It was observed that the specimen without clay was perforated at 101.84 m/s and the specimen with 2% clay was perforated at 122.32 m/s, and the enhancement of the velocity was about 20%. The specimen with 3% clay showed an enhancement of 22% when compared to the specimen without clay. The specimen with 5% clay was perforated at 129.51 m/s, where the enhancement of the velocity was about 27%. It was observed that the enhancement of the ballistic velocity was better for a specimen with clay up to 2%, and thereafter, the enhancement was only marginal. The specimens were tested for the above ballistic impact velocities of 135 m/s, 140 m/s, and 145 m/s. In all the cases, the specimen without clay showed higher residual velocities than the specimen with 1%–5% clay. The decrease in the residual velocity was high for the specimen with 5% clay when compared to the specimen without clay. It means that the specimen with 5% clay has a higher impact resistance capacity. The decrease in the percentage of the residual velocity for specimens with 2%, 3%, 4%, and 5% clay when subjected to an impact velocity of 135 m/s was 35%, 40%, 46%, and 55%, respectively, when compared to specimens without clay. Clay dispersion up to 2% offered high resistance to perforation. This can be observed through the change in the slope shown in Figure 5. When the specimens were subjected to a velocity of 140 m/s, the decrease in the percentage for the specimens with 2%, 3%, 4%, and 5% clay was 29%, 33%, 38%, and 43%, respectively. For the corresponding values for a velocity of 145 m/s, the decrease in the percentage of the residual velocities was 23%, 27%, 30%, and 35%, respectively.

There is a limited amount of research that has focused on developing theoretical models to understand impact phenomena. In this regard, Balaganesan et al. [56] developed a dynamic model to identify the energy dissipated by target specimens in various failure modes and also to predict the ballistic velocity. They used the mass and initial velocity of the projectile and the target’s elastic properties as input parameters for the model. The model also presents the decrease in projectile energy and how the energy is absorbed by specimens through failure modes for every interval of the time period. The failure modes of the specimens elastically deform in the direction of the projectile movement, damage due to delamination, epoxy matrix cracking, and fracture of fibers due to stretching under the point of impact.

From the model, they observed that 70% of the projectile energy was absorbed by the specimens due to elastic deformation of fibers that are not in contact with the projectile. This is due to the higher area of the target that is not in direct contact with the projectile. It was observed that the increase in energy absorption was observed for specimens with clay up to 5%. The authors validated the model results with experiments based on their earlier study.

Projectile energy during perforation and various failure modes of the specimen are shown in Figure 6 for a specimen with 3% clay. The nanomodified specimen absorbs about 50% more energy than the specimen made using neat epoxy. The change in percentage of the energy absorption in various failure modes is marginal when compared to the neat epoxy specimen. The authors observed that the energy absorbed due to elastic deformation in the direction of the projectile movement, damage due to delamination, epoxy matrix cracking, and fracture of fibers is 70.4%, 17.7%, 6.9%, and 3.1%, respectively, of the energy of the projectile before impact. The model predicted that the time taken for complete penetration by the projectile into the target is about 20 µs.

## 4. Self-Healing Polymer Matrix Composites

Core-shell nanofibers containing healing agents have been employed to improve the mechanical properties of laminated composites and to induce self-healing ability in these types of composites [64]. Table 3 summarizes the research on the development of self-healing laminated composites by interleaving core-shell nanofibers. Self-healing laminated polymeric composites using core-shell nanofibers were first developed by Sinha-Ray et al. in 2012 [65]. They encapsulated dicyclopentadiene (DCPD) and isophorone diisocyanate within core-shell nanofibers via the emulsion electrospinning method, while the polyacrylonitrile (PAN) was utilized as the shell material. To prove the healing ability of this core-shell nanofiber, Sinha-Ray et al. incorporated the prepared nanofibers between the third and fourth layers of a six-layer carbon/epoxy composite. The composite was broken, treated with a releasing healing agent, and cured for 24 h. The interlayer containing core-shell nanofibers was investigated by scanning electron microscopy to prove the release of the healing agent and solidification.

In 2013, Wu and his coworker encapsulated DCPD in PAN nanofiber via coaxial electrospinning [66]. The prepared core-shell nanofibers were incorporated between carbon layers in a carbon/epoxy composite, while Grubbs’ catalyst was added to the matrix before composite fabrication. The vacuum-assisted resin transfer molding (VARTM) technique was employed for composite fabrication. Three-point-bending tests were utilized to evaluate the self-healing behavior of the hybrid composites. The fabricated composite showed approximately a 100% stiffness recovery after the breakage and healing process. Scanning electron microscopy (SEM) was also used to investigate the healing in the resin-rich interlayer. The fractographical analyses confirmed the release of the DCPD from ruptured nanofibers and solidification after encountering the Grubbs’ catalyst, which was dispersed in the epoxy matrix. Figure 7 presents the core-shell nanofibers in an epoxy interlayer containing Grubbs’ catalyst. It can be observed that after a breakage in the matrix and subsequently in the composites, the DCPD was released from damaged nanofibers and was solidified by the catalyst, which was dispersed within the epoxy matrix

A comprehensive series of studies on the development of self-healing composites using nanofibers was carried out by Neisiany and his coworker from 2016 to 2018 [67,68,69]. Figure 8 schematically shows the concept of self-healing, in which the core-shell nanofibers are incorporated between the reinforcement layers, breaking the nanofibers, releasing the healing agent, and, consequently, solidifying the healing agent.

Neisiany et al. firstly encapsulated an epoxy and amine-based curing agent within the PAN nanofibers [71] and incorporated them between the carbon layers of carbon/epoxy composites. They evaluated the effect of the nanofiber’s incorporation on the mechanical properties and self-healing behavior of the composite. In the next study, they encapsulated an epoxy and amine-based curing agent within the nanofiber with poly(styrene-co-acrylonitrile) (SAN) as the shell [68]. Their results showed that the use of SAN as the shell provided a higher amount of encapsulation as well as a high encapsulation yield. However, the high amount of healing agent did not improve the self-healing performance in comparison with the PAN nanofiber. Figure 9 shows the morphology of the prepared nanofibers on the surface of carbon fibers and in the carbon/epoxy composites. In their next research, they again encapsulated an epoxy resin and amine-based curing agent in the poly(methyl methacrylate) (PMMA) shell. The hybrid carbon/epoxy composite with this type of core-shell nanofibers showed higher self-healing performance. From the results, Neisiany et al. concluded that for composites reinforced with core-shell nanofibers, the diameter of the prepared nanofibers has a significant effect on improving the out-of-plane mechanical properties of the laminated composites, whereas the mechanical properties (strain at break) of the nanofibers dictate the improvement of the in-plane properties of the composites and self-healing performance. In addition, they claimed the incorporation of the core-shell nanofibers did not have a remarkable effect on the impact properties of the hybrid composites.

Zanjani et al. encapsulated healing agent in the core-shell nanofibers via tri-axial electrospinning [72] and employed them for fabrication of self-healing glass fiber/epoxy composites [70]. They used epoxy resins and its hardener as a healing agent and PMMA as the outer shell. The use of the inner shell extended the lifetime of the healing functionality after encapsulation. Flexural tests besides the fractographical analyses were employed to evaluate the self-healing ability of the hybrid composites. The results confirmed that the incorporation of the prepared tri-axial nanofibers induced the self-healing ability and improved the mechanical properties of the glass fibers/epoxy composites, which are highly prone to the failure.

## 5. Conclusions

There is no doubt that polymer matrix composites are an emerging class of materials for innumerable applications, thus replacing their traditional counterparts. The present paper provided a brief review of the advancements in nanoengineered polymer matrix composites. This review indicates that significant knowledge has been achieved in the area through extensive research conducted by various researchers. The high-velocity impact studies from 2000 till present were reviewed and important events were tabulated. Though laminated polymer matrix composites are already superior in regards to their properties compared with their conventional counterparts, they are further enhanced by the addition of or modification with other materials. In this review, the nanofillers and nanofiber interleaved composites were introduced and reviewed. More importance is given to the high-velocity impact response of nanofiber interleaved composites. Self-healing composites, via the interleaving of nanofibers that encapsulate healing agents, were reviewed as the technique provides solutions for a long standing concern of composite repair. Though various reviews have been published in the considered topic, there remains a significant gap in information regarding the use of nanoparticles and nanofibers while considering their effect on composite laminates. The study of high velocity impacts on nanofiber reinforced composites is another untouched area. This paper described various studies on nanoparticle modified composites, but various issues, like the agglomeration of these particles, uniform distribution during fabrication, the alignment of nanoparticles, etc., need to be addressed. The core-shell approach has gained popularity over the last few years and more applications in various other fields are currently being tested. From this review, we can conclude that increasingly more research is being carried out on the incorporation of nano level particles and materials to improve the properties of fiber reinforced polymer matrix composites. Furthermore, significant scope for further development still exists. This paper provides insight and a brief review of the state-of-the-art in nano-engineered polymer matrix composites.

## Figures and Tables

**Figure 1 polymers-11-00644-f001:**
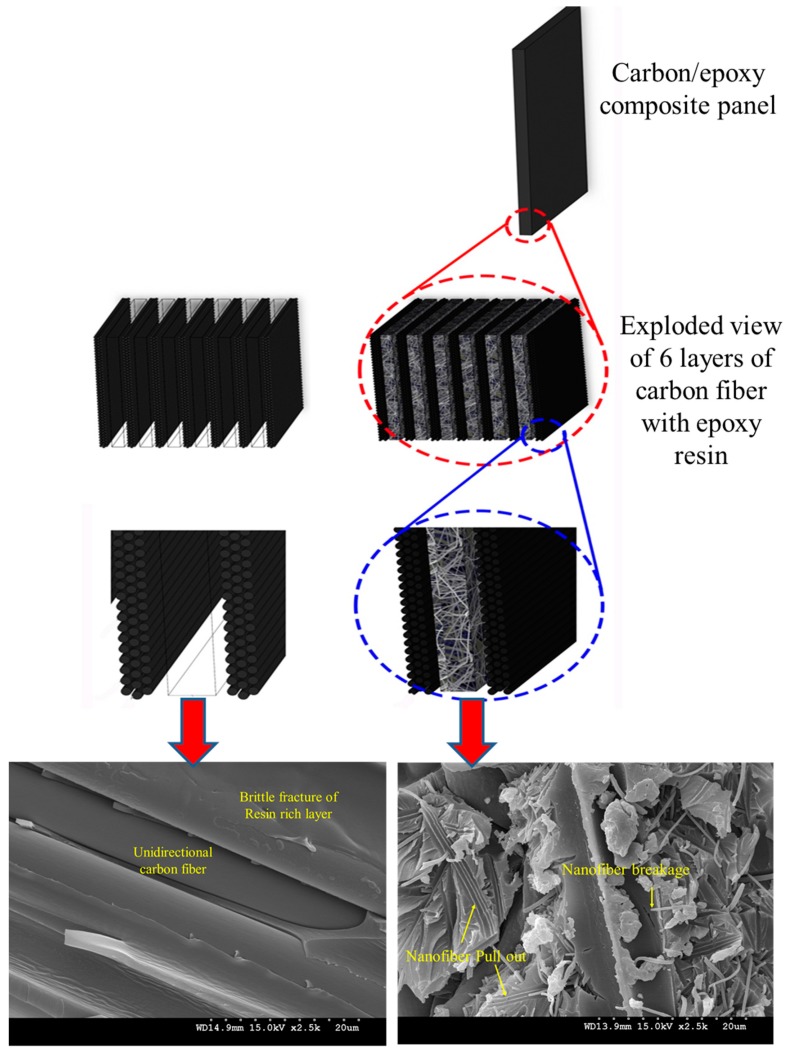
Schematic illustration of the laminated composite with and without interleaved nanofibers. Adapted with permission from [42], and FESEM micrographs of the cross-section of a carbon/epoxy composite with (right) and without interleaved nanofibers (left). Adapted with permission from [41].

**Figure 2 polymers-11-00644-f002:**
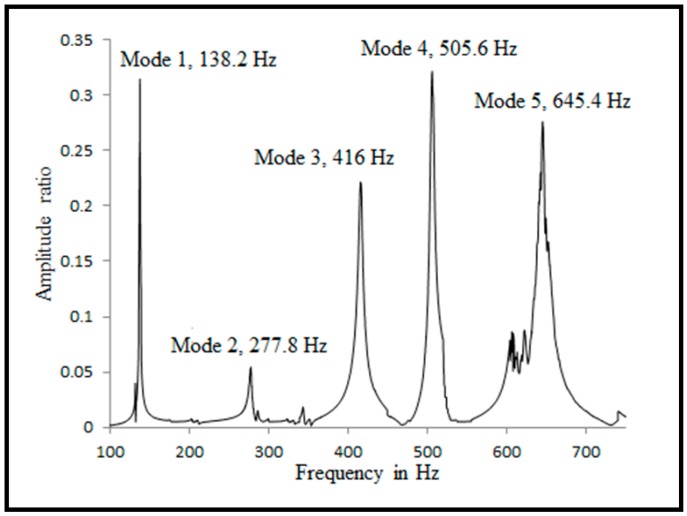
FRF plot for nanocomposite laminate with 3% clay.

**Figure 3 polymers-11-00644-f003:**
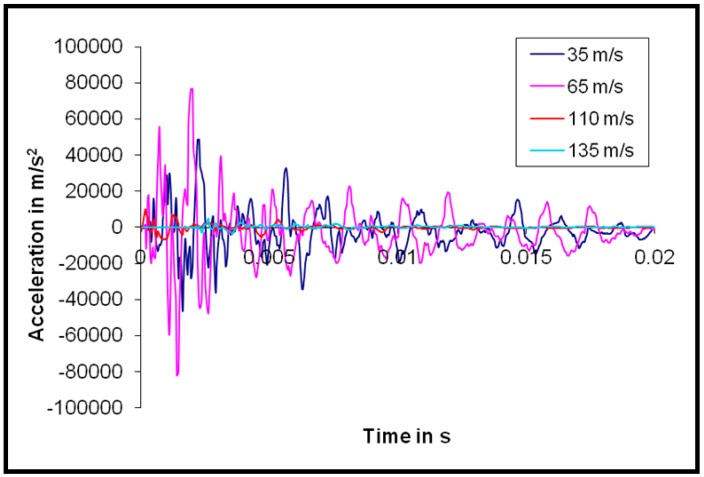
Acceleration-time response for nanocomposite laminate with 3% clay for velocities below and above the ballistic limit. Reprinted with permission from [54].

**Figure 4 polymers-11-00644-f004:**
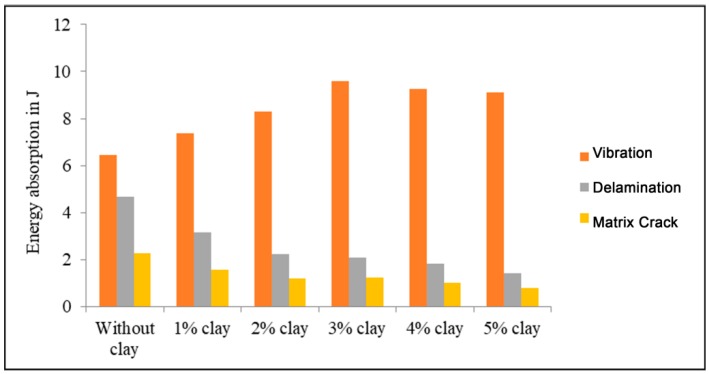
Energy absorbed by three-layer laminates when subjected to an impact velocity of 82 m/s.

**Figure 5 polymers-11-00644-f005:**
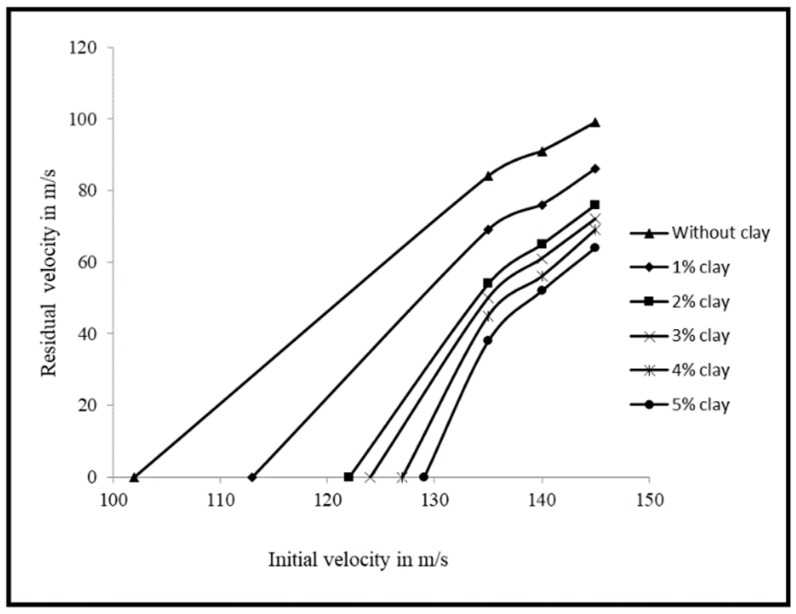
Initial velocity vs residual velocity for nanocomposite specimens. Reprinted with permission from [56].

**Figure 6 polymers-11-00644-f006:**
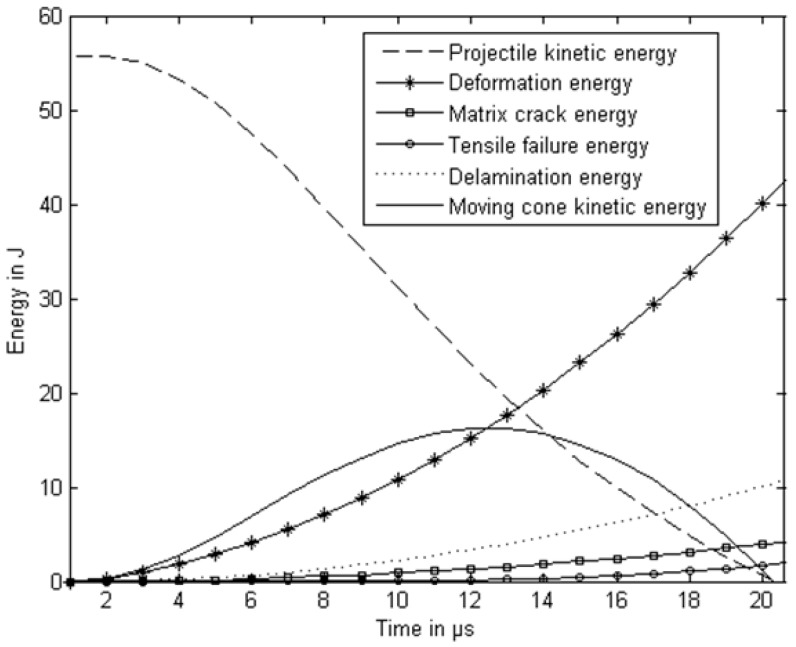
Different modes of energy for a three-layer specimen with 3% clay when subjected to 122.3 m/s (ballistic limit). Reprinted with permission from [55].

**Figure 7 polymers-11-00644-f007:**
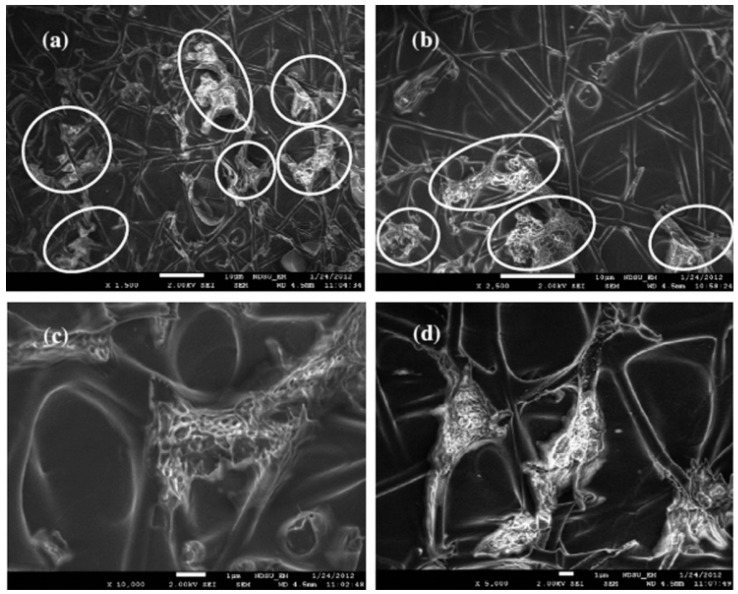
SEM images of the fracture surfaces of the self-healing carbon/epoxy composite containing core-shell nanofibers after a three-point bending test. (**a**,**b**) Core-shell nanofibers in the resin rich layer (circled spots showed the released DCPD damage and solidification). (**c**,**d**) Solidification of DCPD after encountering the catalyst. Reprinted with permission from [66].

**Figure 8 polymers-11-00644-f008:**
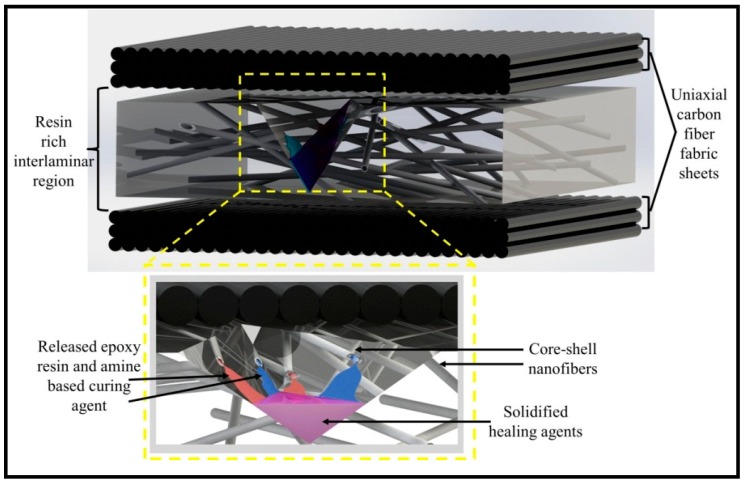
The concept of the self-healing process in laminated composites using core-shell nanofibers. Reprinted with permission from [69].

**Figure 9 polymers-11-00644-f009:**
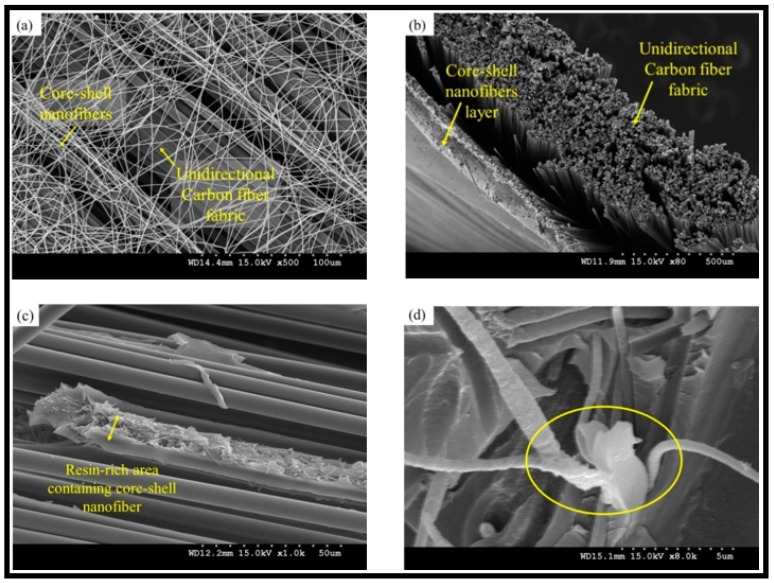
SEM images of the core-shell nanofibers on the surfaces of unidirectional carbon fibers. (**a**) After deposition of 0.05 g m^−2^; (**b**) after deposition of 1 g m^−2^; (**c**) SEM image of the hybrid composite cross-section after the tensile test; (**d**) the release of healing agents from ruptured nanofibers and solidification in the composite. Reprinted with permission from [68].

**Table 1 polymers-11-00644-t001:** Velocity ranges and their applications [29]

Velocity	Impact Testing Equipment	Material Test Method	Applications
Low (0–50 m/s)	Drop hammer Pneumatic accelerator	Hydraulic Servo-hydraulic Screw-driven	Dropped objects Vehicle impact/ship collision Crash-worthiness of containers for hazardous materials
Sub-ordnance (50–500 m/s)	Compressed air gun Gas gun	Pneumatic Hydraulic Taylor impact tests Split Hopkinson Pressure bar (SHPB)/Tension bar (SHTB)	Design of nuclear containment Free-falling bombs and missiles Fragments due to accidental explosions
Ordnance (500–1300 m/s)	Compressed air gun Gas gun	Taylor tests SHPB/SHTB	Military
Ultra-ordnance (1300–3000 m/s)	Powder gun 2-Stage light gas gun	Taylor impact tests	Military
Hypervelocity (>3000 m/s)	2-Stage light gas gun	Taylor impact tests	Space Vessels Exposed to meteoroid impact and space debris

**Table 2 polymers-11-00644-t002:** Property enhancement due to nylon nanofibrous interleaving [52].

Nano-Fibers	Diameter (nm)	Matrix	Properties: Value of Matrix/Value of Composites; % Absolute Increased Properties
Nylon-4,6	30–200	Epoxy	Transparent; Young’s modulus: 2.5/91 MPa; fracture stress: 1.82/2.4 MPa
Gr-nylon-6 (Gr 0.01 wt.%)	300–500	PMMA	Transparent; tensile strength: 56%; modulus: 113%; toughness: 250%
Nylon-6	200–400	PMMA	Transparent; bending strength: 12%; bending modulus: 30%; tensile strength: 20%; tensile modulus: 32%
Nylon-6	200–400	PMMA	Transparent; tensile strength and modulus: >20%
Nylon-6	134	PVA	Tensile failure stress: 34/740 MPa; tensile failure strain: 340%/490%
Nylon-6	100–600	BIS-GMA/TEGDMA	Flexural strength: 36%; elastic modulus: 26%; work of fracture: 42%
Nylon-6/silica nanocrystal	250	BIS-GMA/TEGDMA	Flexural strength: 23%; elastic modulus: 25%; work of fracture: 98%
Nylon-6,6	150	Epoxy	Mechanical energy absorbing capability: 23.2%; maximum stress: 6.5%
Nylon-6,6	75–250	Epoxy	Impact force: 900/2100 N; impact energy: 0.46/1.8 J; impact damage growth rate: 0.115/0.105 mm^2^ N^−1^
Nylon-6,6	150–300	Carbon/epoxy	Fracture toughness: 156% (mode I) and 69% (mode II)
Nylon-6	(A) 150	Glass fiber/epoxy	(A) Stress: 550/581 MPa, shear modulus: 4.0/4.7 GPa
	(B) 230		(B) Stress: 550/611 MPa, shear modulus: 4.0/4.7 GP; GIC: 1264/1447 J m^−2^
Nylon-6,6	150	Glass/epoxy	Energy release rate GI: 4.5%; GII: 68%
Nylon-6	800	PCL	Young’s modulus: 352/530 MPa; stress at break: 15.9/18.1 MPa; strain at break: 467%/601%
Nylon-6	800	PLA	Modulus: 2.4/6.6 GPa; stress at break: 48/46 MPa; strain at break: 3.6%/1.7%
Nylon-6	220	Melamine-formaldehyde	(A) Stress: 47.5/74.5 MPa; strain: 76.2%/2.85%; modulus: 0.37/2.88 GPa; toughness: 21.8/1.0 J g^−1^
			(B) Stress: 47.5/77.9 MPa; strain: 76.2%/38.4%; modulus: 0.37/0.85 GPa; toughness: 21.8/17.6 J g^−1^
Nylon-6	150–300	TPU	Transparent; stress: 42.27/82.98 MPa; strain: 672.9%/876.0%; modulus: 27.1/ 51.9 MPa; toughness: 108.47/274.83 J g^−1^
Nylon-6,6	200–350	Polyethylene	Tensile strength: 27.74/32.56 MPa; elongation: 1184%/1341%; toughness: 249.36/315.07 MJ m^−3^
Nylon-6,6	150	Gr-epoxy	Mechanical energy absorbing capability: 23.2%; GIC: 5%
Nylon-6,6/GNPs	90–150	Aramid/epoxy	Elongation at break: 19.9%/34.48%; toughness: 68%
Nylon-6	200–400	PMMA	Transparent; tensile strength: 27.5/54.4 MPa; modulus: 0.61/1.12 GPa; toughness: 0.7/2.1 MJ m^−3^
Nylon-6,6	300	Cyclic butylene terephthalate	Transparent; stress: 25/44 MPa
Nylon-6	100	Protein	Tensile strength: 0.024/0.136 MPa; elastic modulus: 1.57/1.8 GPa

**Table 3 polymers-11-00644-t003:** List of the recent progress in the development of self-healing laminated composites by interleaving of core-shell nanofibers.

Composite	Method of Encapsulation	Healing Agent	Shell Material	Methods of Self-Healing Evaluation	Reference
Carbon/epoxy	Emulsion electrospinning	dicyclopentadiene and isophorone diisocyanate	Polyacrylonitrile	Fractographical analysis	[65]
Carbon/epoxy	Coaxial electrospinning	dicyclopentadiene	Polyacrylonitrile	Mechanical tests and Fractographical analysis	[66]
Carbon/epoxy	Coaxial electrospinning	Low viscosity epoxy resin and amine-based curing agent	Polyacrylonitrile	Mechanical tests and Fractographical analysis	[67]
Carbon/epoxy	Coaxial electrospinning	Low viscosity epoxy resin and amine-based curing agent	poly(styrene-co-acrylonitrile)	Mechanical tests, heat of healing reaction, and Fractographical analysis	[68]
Carbon/epoxy	Coaxial electrospinning	Low viscosity epoxy resin and amine-based curing agent	poly(methyl methacrylate)	Mechanical tests and Fractographical analysis	[69]
Glass fiber/epoxy	Tri-axial electrospinning	epoxy resin and its curing agent	poly(methyl methacrylate) as outer shell and polyacrylamide as middle wall	Mechanical tests and Fractographical analysis	[70]

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
