# Peer review of "A Review of Recent Advances in Nanoengineered Polymer Composites"

_polymers, 2019, doi:10.3390/polym11040644_

Round 1
Reviewer 1 Report
Dear Authors, I have read your manuscript and some questions raised. Attached please find my notes. Overall. Interesting review about nanoengineered polymer composites. Introduction: Authors stated that "The fibre diameters may range from a few millimeters to sub nanometric ranges". Please add a reference for this statement. Introduction. Authors stated that " These fibre reinforced composites finds innumerous applications owing to their superiority in various properties". Authors could be more detailed. Authors could stress the concept of the advantages of fiber reinforced composites over unreinforced ones. Therefore, when introducing FRCs, Authors could add that " fiber reinforced composites FRCs are axial particulates embedded in fitting matrices. The advantages of FRCs over unreinforced ones are well known and their characteristics are useful for many scopes. The fields with a constantly increasing number of studies about FRC applications are Construction Science (Long-Term Flexural Behaviors of GFRP Reinforced Concrete Beams Exposed to Accelerated Aging Exposure Conditions. Yeonho Park, Young Hoon Kim , and Swoo-Heon Lee. Polymers 2014, 6(6), 1773-1793;) Aerospace (Fibre reinforced composites in aircraft construction. Soutis C. Progress in Aerospace Sciences Volume 41, Issue 2, February 2005, Pages 143-151), orthopedics (Use of carbon-fiber-reinforced composite implants in orthopedic surgery. Hak DJ, Mauffrey C, Seligson D, Lindeque B. Orthopedics. 2014 Dec;37(12):825-30.) and Dentistry (Fiber-Reinforced Composites for Dental Applications. Scribante A, Vallittu PK, Özcan M. Biomed Res Int. 2018 Nov 1;2018:4734986.).” These considerations could be added in Introduction Section. Introduction: Authors stated that " The dynamic response of these materials under various loading conditions were studied to check their suitability for certain applications". Please add a reference for this statement. 1.1Impact on Composites. Authors stated that “The impact studies on composites laminates have always been an area of concern for various researchers globally". In fact, Authors refer to various studies but in the initial part of the section there are only 2 references (3 and 4). Please add some references to sustain this statement. For example, when saying “In general, there are five types of velocity considered for the studying of impact responses: low velocity, sub ordnance, ordnance, ultra-ordnance, and hypervelocity” it could be useful to have a reference for each different velocity. 1.2State-Of-The-Art in Composite Impact Research. Authors stated that "The dynamic response of these materials under various loading conditions were studied to check their suitability for certain applications". Please add databases used (Scopus, WoS…). It is very important to define on which database the research has been conducted. 2.1Electrospun Nanofiber Mats. Authors stated that " Electrospinning is one of the most commonly used methods to produce nanofibres ranging from several micro meters to sub nanometric ranges". Please add a reference for this statement. Graphs. All the graphs are under permissions of Authors reported in references 66 (Balaganesan, G.; Velmurugan)67 (Balaganesan, G.; Velmurugan, R.; Kanny, K), 68 (Balaganesan, G.; Velmurugan, R.; Srinivasan, M.; Gupta, N.; Kanny, K), , 69 (Velmurugan, R.; Balaganesan, G.), 76 (Neisiany, R.E.; Lee, J.K.Y.; Khorasani, S.N.; Ramakrishna, S.), 77 (Esmaeely Neisiany, R.; Lee, J.K.Y.; Nouri Khorasani, S.; Bagheri, R.; Ramakrishna, S). All these graphs are from the same two research groups. If it is possible, in my opinion it would be better to report also data from other research groups. 3. Nanofiller Reinforced Polymer Matrix Composites. It has been stated that “The authors have conducted experiments for various initial velocities of projectile and have obtained residual velocities for each case of initial velocity”. Please add also other Literature 3. Nanofiller Reinforced Polymer Matrix Composites. The chapter 3 (6 pages) is unbalanced with the other three chapters 1, 2 and 4 (1, 2 and 3 pages respectively) that are much shorter. In my opinion, it would be better to add some discussion in other sections or reduce section 3. Photographs: ok Tables: ok References: some literature seems to be not indexed in Scopus. Please see reference number 8,30, 48,49, 57. As the text is a Review, where possible, it would be better to have Scopus indexed References.
Author Response
Title of the paper: A Review of Recent Advances in Nanoengineered Polymer Composites
Reviewer 1:
General remark:
I have read your manuscript and some questions raised. Attached please find my notes. Overall. Interesting review about nanoengineered polymer composites.
Comment 1:
Introduction: Authors stated that "The fibre diameters may range from a few millimeters to sub nanometric ranges". Please add a reference for this statement.
Answer:
The authors would like to appreciate the esteemed reviewer for the time and insightful comments provided. The reference has been added in the revised manuscript as suggested.
Comment 2:
Introduction. Authors stated that " These fibre reinforced composites finds innumerous applications owing to their superiority in various properties". Authors could be more detailed.
Answer:
This is addressed in the revised manuscript as suggested.
Comment 3:
Authors could stress the concept of the advantages of fiber reinforced composites over unreinforced ones. Therefore, when introducing FRCs, Authors could add that " fiber reinforced composites FRCs are axial particulates embedded in fitting matrices. The advantages of FRCs over unreinforced ones are well known and their characteristics are useful for many scopes. The fields with a constantly increasing number of studies about FRC applications are Construction Science (Long-Term Flexural Behaviors of GFRP Reinforced Concrete Beams Exposed to Accelerated Aging Exposure Conditions. Yeonho Park, Young Hoon Kim , and Swoo-Heon Lee. Polymers 2014, 6(6), 1773-1793;) Aerospace (Fibre reinforced composites in aircraft construction. Soutis C. Progress in Aerospace Sciences Volume 41, Issue 2, February 2005, Pages 143-151), orthopedics (Use of carbon-fiber-reinforced composite implants in orthopedic surgery. Hak DJ, Mauffrey C, Seligson D, Lindeque B. Orthopedics. 2014 Dec;37(12):825-30.) and Dentistry (Fiber-Reinforced Composites for Dental Applications. Scribante A, Vallittu PK, Özcan M. Biomed Res Int. 2018 Nov 1;2018:4734986.).” These considerations could be added in Introduction Section.
Answer:
The authors would like to thank the respected reviewer for valuable comments. The application field and relevant references have been cited in the revised version of the manuscript as suggested. The authors believe that the suggested references can be valuable for the readers who follow this type of work.
Comment 4:
Introduction: Authors stated that " The dynamic response of these materials under various loading conditions were studied to check their suitability for certain applications". Please add a reference for this statement.
Answer:
The reference was cited in the revised manuscript.
Comment 5:
1.1Impact on Composites. Authors stated that “The impact studies on composites laminates have always been an area of concern for various researchers globally". In fact, Authors refer to various studies but in the initial part of the section there are only 2 references (3 and 4). Please add some references to sustain this statement.
Answer:
Additional references have been included in the revised manuscript.
Comment 6:
For example, when saying “In general, there are five types of velocity considered for the studying of impact responses: low velocity, sub ordnance, ordnance, ultra-ordnance, and hypervelocity” it could be useful to have a reference for each different velocity.
Answer:
The reference for each velocity has been cited in the revised manuscript.
Comment 7:
1.2State-Of-The-Art in Composite Impact Research. Authors stated that "The dynamic response of these materials under various loading conditions were studied to check their suitability for certain applications". Please add databases used (Scopus, WoS…). It is very important to define on which database the research has been conducted.
Answer:
The database for the research has been included in the revised version of the manuscript
Comment 8:
2.1Electrospun Nanofiber Mats. Authors stated that " Electrospinning is one of the most commonly used methods to produce nanofibres ranging from several micro meters to sub nanometric ranges". Please add a reference for this statement.
Answer 8:
The relevant reference has been included in the revised manuscript.
Comment 9:
Graphs.All the graphs are under permissions of Authors reported in references 66 (Balaganesan, G.; Velmurugan)67 (Balaganesan, G.; Velmurugan, R.; Kanny, K), 68 (Balaganesan, G.; Velmurugan, R.; Srinivasan, M.; Gupta, N.; Kanny, K), , 69 (Velmurugan, R.; Balaganesan, G.), 76 (Neisiany, R.E.; Lee, J.K.Y.; Khorasani, S.N.; Ramakrishna, S.), 77 (Esmaeely Neisiany, R.; Lee, J.K.Y.; Nouri Khorasani, S.; Bagheri, R.; Ramakrishna, S). All these graphs are from the same two research groups. If it is possible, in my opinion it would be better to report also data from other research groups.
Answer:
The authors would like to appreciate the esteemed reviewer for the insightful comment. Some additional references and more figures have included in the revised manuscript.
Comment 10:
3. Nanofiller Reinforced Polymer Matrix Composites. It has been stated that “The authors have conducted experiments for various initial velocities of projectile and have obtained residual velocities for each case of initial velocity”. Please add also other Literature 3. Nanofiller Reinforced Polymer Matrix Composites.
Answer:
Few more literature has been included in the revised manuscript.
Comment 11:
The chapter 3 (6 pages) is unbalanced with the other three chapters 1, 2 and 4 (1, 2 and 3 pages respectively) that are much shorter. In my opinion, it would be better to add some discussion in other sections or reduce section 3.
Answer:
Chapters 1, 2 and 4 have been added and reduced in Section 3 as suggested by the respected reviewer.
Comment 12:
Photographs: ok Tables: ok References: some literature seems to be not indexed in Scopus.
Answer:
The authors would appreciate the esteemed reviewer for his concerns about the Tables, Photographs, and References. The references which are not indexed Scopus have not been included in the revised manuscript.
Comment 13:
Please see reference number 8,30, 48,49, 57. As the text is a Review, where possible, it would be better to have Scopus indexed References.
Answer:
References 8,30,48,49 and 57 of the previous version manuscript were verified. The references which are not indexed Scopus have not been included in the revised manuscript.
Once again, the authors extend their thanks to the esteemed reviewer for his valuable comments and suggestions. We’ve tried our best in revising the manuscript as per the suggestions.

Reviewer 2 Report
Dear Authors,
The manuscript is a good effort at summarizing he various researches and findings mainly in the field of the laminated polymer matrix composites. However, it seems authors have mainly summarized the key finding from various publications in the field by various authors, but no effort is made to pull together a comprehensive and clear story, to bring all of it together. This needs to rectified in the revised manuscript.
Additionally,
1) There are numerous acronyms throughout the manuscript not defined before 1st use such as DF, NF, FRF and various other polymer names.
2) Also, there are many typos and grammatical mistakes throughout the manuscript.
line 122-123 - font used is different
line 298 - the specimen without clay
line 380 - English grammer
line 402 - typo
3) Authors have used the term specimend throughout manuscript - never heard of it.
Thanks
Author Response
Title of the paper: A Review of Recent Advances in Nanoengineered Polymer Composites
Reviewer 2:
General remarks:
The manuscript is a good effort at summarizing the various researches and findings mainly in the field of the laminated polymer matrix composites. However, it seems authors have mainly summarized the key finding from various publications in the field by various authors, but no effort is made to pull together a comprehensive and clear story, to bring all of it together. This needs to rectified in the revised manuscript.
Answer:
The authors would like to appreciate the esteemed reviewer for the insightful comment, the manuscript was double checked and accordingly edited to avoid any grammatical mistakes. Furthermore, the revised manuscript addresses the above remarks.
Comment 1:
1) There are numerous acronyms throughout the manuscript not defined before 1st use such as DF, NF, FRF and various other polymer names.
Answer:
The acronyms are defined before their use in the revised manuscript.
Comment 2:
2) Also, there are many typos and grammatical mistakes throughout the manuscript.
line 122-123 - font used is different
line 298 - the specimen without clay
line 380 - English grammar
line 402 - typo
Answer:
The above sections were corrected, and the revised manuscript is thoroughly checked for typo and grammatical errors.
Comment 3:
3) Authors have used the term specimend throughout manuscript - never heard of it.
Answer:
The authors would humbly thank the reviewer for his comment and would like to add that, it is mentioned in section 4 in the previous version of the paper. Now corrected as 'laminated' in the revised manuscript.
Once again, the authors extend their thanks to the esteemed reviewer for his valuable comments and suggestions. We’ve tried our best in revising the manuscript as per the suggestions.
Round 2
Reviewer 1 Report
Dear Authors,
You have corrected the manuscript following all comments.
Good job
Author Response
The authors would like to appreciate the esteemed reviewer for the time and insightful comments provided. The manuscript has been thoroughly checked and corrected for grammatical mistakes and other language issues.
Reviewer 2 Report
Authors have addressed the concerns raised earlier in the revised manuscript. English language check is still required as some sentences are still grammatically incorrect and words are capitalized when they shouldn't.
example : page 1 - There has been researching, to further enhance.....
Author Response

(The authors gave the same response as above.)
